# Prevalence and health consequences of nonmedical use of tramadol in Africa: A systematic scoping review

**Saidou Sabi Boun**[1], **Olumuyiwa Omonaiye**[2,3], **Sanni Yaya**[1,4]*

**1** School of International Development and Global Studies, University of Ottawa, Ottawa, Ontario, Canada, **2** Centre for Quality and Patient Safety Research, Institute for Health Transformation, Deakin University, Geelong, Australia, **3** Deakin University Centre for Quality and Patient Safety Research–Eastern Health Partnership, Box Hill, Victoria, Australia, **4** The George Institute for Global Health, Imperial College London, London, United Kingdom

* sanni.yaya@gmail.com

**Data Availability Statement:** All relevant data and related metadata underlying the findings reported in the paper are available in the Supporting Information files.

## Abstract

Tramadol is a widely prescribed painkiller around the world. As a synthetic opioid, it offers a valuable substitute for morphine and its derivatives in African countries. However, the adverse health effects of tramadol use resulting from illicit trafficking, like those caused by fentanyl and methadone in North America, have not been well-documented in Africa. This scoping review aims to shed light on the nature and scope of the nonmedical use (NMU) of tramadol in Africa and its associated health consequences. To carry out our scoping review, we used Arksey and O'Malley's (2005) five-step approach for exploratory analysis and followed Joanna Briggs Institute guidelines for scoping reviews to ensure systematic and replicable studies. We then searched six databases: Medline, Global Health (EBSCO), Scopus, Web of Science, the African Journals online database, and for grey literature via Google Scholar without any time restriction. The articles were imported into Covidence and reviewed by two independent researchers. Eighty-three studies on NMU of tramadol's prevalence or health consequences were selected from 532 titles/abstracts screened, including 60 cross-sectional and six qualitative studies from 10 African countries. Findings from the included studies highlighted five distinct groups significantly affected by the NMU of tramadol. These groups include: 1) young adults/active populations with varying degrees of prevalence ranging from 1.9% to 77.04%, 2) professionals, where drivers exhibit a relatively high prevalence of tramadol NMU, ranging from 7.2% to 35.1%, and commercial motorcyclists, with a prevalence of 76%, 3) patients, who have a high rate of tramadol NMUs, with prevalence rates ranging from 77.1% to 92%, 4) academics, with a considerable rate of tramadol misuse among substance-using undergraduates (74.2%) and substance-using high school students (83.3%), and 5) other individuals impacted in various ways. The health consequences are classified into four distinct types: intoxication, dependence syndrome, withdrawal syndrome and other symptoms. Despite providing a comprehensive global overview of the phenomenon described in the African literature, this systematic scoping review's main limitations stem from the relatively limited exploration of various consequences of the NMU of tramadol, notably those of a social and economic nature. Our review shows that tramadol

**Funding:** The authors received no specific funding for this work.

**Competing interests:** The authors have declared that no competing interests exist.

misuse affects diverse populations in Africa. The prevalence of misuse varies within sub-populations, indicating the complexity of the issue. Professional and academic groups have different rates of misuse across regions. This highlights the need for targeted interventions to address unique challenges contributing to tramadol misuse. Future studies should focus on the social and economic costs of abuse on households to better understand the impact on well-being.

**Systematic review registration:** Open Science Framework: https://osf.io/ykt25/.

## Introduction

Tramadol is among the most prescribed pain relievers on a global scale [1]. This synthetic opioid is utilized for treating moderate to severe pain and offers a viable alternative to morphine, owing to its reliable accessibility and low cost [2]. In its early evaluation, it was assessed as having common addictive properties, so it was not placed under international control, like fentanyl and methadone, which are responsible for the opioid crisis in US and Canada. However, according to United Nations Office on Drugs and Crime (UNODC), the number of opioid users has risen from 31 million in 2010 to 61 million in 2020, including 1.2% of nonmedical users (including tramadol nonmedical users) aged between 15 and 64. Additionally, opioid use is the cause of 77% of drug-related deaths and accounts for 13 million disability-adjusted life years lost in 2020 [3]. The nonmedical use (NMU) of drug refers to using it in ways not prescribed by a healthcare professional, such as different dosages, frequencies, or purposes than intended.

Tramadol was first synthesized by *Grünenthal* Pharmaceutical Company in 1962. It is a synthetic codeine analog with a dual mechanism of action, affecting μ-opioid receptors and inhibiting serotonin and noradrenaline reuptake [4]. The liver metabolizes it through cytochrome P450, particularly CYP2B6, producing two metabolites, M1 and M2, that account for most of its pain-relieving activity [5]. M1, the metabolite, has a 200–300 times greater affinity for μ-opioid receptors than the parent compound and double the analgesic potency [4]. Tramadol is quickly absorbed after administration, and extended-release tablets can release their active ingredient for up to 12 hours, reaching peak concentrations around 5 hours after consumption. Its bioavailability is around 95%, with an elimination half-life of about 6 hours [5].

In Africa, tramadol is widely prescribed for neuropathic and nociceptive pain, back pain, and people with cancer and is often used for the pain of sickle cell anemia [6, 7]. A study was carried out to investigate the inclusion of tramadol in the national essential drug lists of 112 documents published between 2002 and 2014, spanning 39 African, 23 American, 30 Asian (including Middle Eastern), 8 European, and 12 Oceanic countries. The results indicated that tramadol was featured as the sole second-line analgesic agent in almost half of the countries surveyed [8]. Many African countries have included tramadol in their national essential medicine lists. These countries are Algeria (2007), Botswana (2012), Congo (2013), Ivory Coast, Egypt (2006), Ethiopia (2015), Ghana (2010), Morocco (2008), Namibia (2008), Rwanda (2010), South Africa (2006), Sudan (2010), Togo (2012), Tunisia (2008), and Tanzania (2013) [1]. However, among these countries, only Egypt has implemented strict national control over tramadol. Additionally, several other countries around the world, including Bahrain since 2000, Australia since 2001, Sweden since 2008, and Iran since 2007, have also placed tramadol under national control [1].

Multiple research studies have shown an increase of NMU of tramadol in Africa, resulting in addiction symptoms comparable to morphine. This is particularly notable when the

substance is consumed consistently in quantities surpassing the suggested therapeutic range [4, 9–14]. As a result, over several years, tramadol has evolved into a significant public health concern in Africa, paralleling the emergence of fentanyl and methadone-related issues in North America. There has been a significant increase in the illicit use of tramadol among younger populations in African countries, as indicated by the authorities' confiscation of large amounts of this substance in recent years. For instance, 17 tonnes of smuggled tramadol were seized in West African countries in 2014, 121 tonnes in 2015, and a staggering 170 tonnes in 2017 [2]. The NMU of tramadol is even more harmful as the illicit doses sold on the pavements and markets are 2 to 5 times higher than the usual doses (100 to 250 mg against 50 mg usually), thus increasing its addictive power [13, 14]. Tramadol's affordability, accessibility, and ease of concealment make it popular among young Africans [2]. This explains why it is described as the cocaine of the poor' [14]. In most African countries, it is easy to obtain in pharmacies without a medical prescription [2]. In North African countries, a literature review showed that tramadol is the second most used drug by students in Egypt due to its psychoactive properties [15] and more generally in countries in the Eastern Mediterranean region, including Libya, Morocco, Somalia and Tunisia [16]. In Central Africa, especially Cameroon, tramadol is used illicitly for its psychoactive properties and to increase work efficiency [17]. Tramadol is marketed as a standalone product or in combination with paracetamol under various brand names including Ixprim, Ultram, Trabar, Zamadol, and Zamudol [1].

In several African countries, tramadol is commonly ingested alongside tea, coffee, or alcohol and combined with other pharmaceutical drugs like benzodiazepines [12, 13]. It is consumed collectively during weddings, baptisms, and other community activities and is occasionally blended with cannabis to seek a euphoric sensation, a primary motive behind its use [18]. Additionally, individual consumption is observed, where manual labourers employ it to combat fatigue or enhance physical and sexual performance [18, 19]. In Cameroon, farmers take large amounts of tramadol and give some to cattle to plough the soil longer [19]. The utilization of tramadol is frequently linked to public disturbances and traffic accidents due to potential side effects such as dizziness, euphoria, and alterations in fear and pain perception [18, 20]. The escalating NMU of tramadol across many African nations has led some experts to characterize the situation as an opioid crisis attributed to tramadol [21]. In 2017, the UNODC issued a warning regarding the adverse impact of NMU and the trafficking of tramadol on the economies and security of the entire Sahel and Niger Delta region, emphasizing its potential social consequences [19].

The WHO Expert Committee on Drug Dependence (ECDD) reviewed tramadol five times in 1992, 2000, 2002, 2006, and 2014 [4]. During the 41st meeting in 2019, although there were concerns raised regarding the growing instances of tramadol abuse in various regions, particularly in low and middle-income countries, and despite the seriousness of the issue, the committee underscored that the existing evidence did not justify the inclusion of tramadol in the list of internationally controlled drugs [4]. This decision was motivated by the lack of access to alternative analgesics in several countries, where tramadol is often the only treatment for moderate to severe pain. Similarly expressing this viewpoint, some authors noted that some parts of Africa, such as West Africa, are confronted with dual opioid challenges: a critical deficiency of opioid analgesics within the healthcare infrastructure, contributing to a substantial load of untreated pain, and a surge of inferior and counterfeit tramadol medications culminating in elevated unauthorized usage and undocumented fatalities [2].

Despite the numerous warnings and media reports about the health and social implications of NMU of tramadol [22–25], there is still a pressing need for scientific research to fully comprehend the impact of this phenomenon on health. Other regions of the world have already undertaken extensive research to examine tramadol abuse among diverse communities,

particularly in Europe, the Middle East, and Iran [26, 27]. However, to the best of our knowledge, no one has yet compiled the available information on the frequency and harmful health effects of NMU of tramadol in African countries. Therefore, this study aims to systematically review the literature to understand the nature and extent of NMU of tramadol use and its health consequences in Africa to guide future research.

## Methods

### Study design

The study protocol outlining the methodology for the design and conduct of the scoping review was prespecified and prospectively registered in Open Science Framework and published [28]. Our study aims to investigate the prevalence and health consequences of NMU of tramadol in African countries. For this research, the NMU of tramadol refers to using it in ways not prescribed by a healthcare professional, such as different dosages, frequencies, or purposes than intended.

We used Arksey and O'Malley's (2005) five-step approach for exploratory analysis and followed Joanna Briggs Institute guidelines for scoping reviews to ensure systematic and replicable studies [29]. The five steps are: 1) formulate the research question; 2) identify relevant studies; 3) select studies according to inclusion and exclusion criteria; 4) extract and map the results; 5) report the results [30]. We reported and filtered our results following the PRIMA-SCR checklist (Preferred Reporting Items for Systematic Reviews and Meta-Analysis Extension for Scoping Reviews) [31].

**Stage 1: Identification of the research questions.** To guide our scoping study, we have developed two research questions as follows:

1.  What is the prevalence of NMU of tramadol in African countries, and what are the characteristics of the populations involved?

2.  What are the health consequences and risk factors associated with NMU of tramadol in African countries?

**Stage 2: Identifying relevant studies.** This scoping review was conducted using PICOS (Population, Interventions, Comparator, Outcome, Setting) framework as recommended by JBI [32]. The five elements of the PICOS framework are reported in S1 Table. The search strategy was straightforward, given the supposed weakness of the African literature about interest. The keyword "tramadol," without any time restriction, added Medical Subject Heading (MeSH) as "Drug abuse," "illicit drugs," or "Prescription Drug Misuse," and "Africa" were used to query the selected databases. These keywords combined with the Boolean operators (and, or, not) resulted in search equations that formed the basis of the search strategy. Our research strategy is detailed in S2 Table.

**Stage 3: Selection of eligible studies.** All articles related to our research question and framework, meeting inclusion criteria and available in English or French in selected databases were included. Six databases were explored: Medline, Web of Science, Scopus, African Journal online database, Global Health (EBSCO) and the grey literature with Google Scholar. Our study included all research conducted in Africa on the prevalence of NMU of tramadol in various population groups, evidence of addiction, intoxication, seizures, and mortality related to NMU of tramadol in various formats. The search was carried out in the above-mentioned databases from January to April 2023.

All 722 articles from our search were uploaded to the Covidence platform, streamlining our screening process and removing duplicates [33]. After eliminating 189 duplicates, 532 articles

underwent a double-blind review of their titles and abstracts by two independent researchers. Subsequently, 132 articles underwent a comprehensive review stage, during which 49 were excluded for various reasons (Fig 1). Any discrepancies regarding the inclusion or exclusion of a study were resolved through consensus, and in cases of unresolved disagreements, a third researcher was consulted. The exclusion criteria were: 1) studies without a specific indicator (prevalence, mortality, morbidity); 2) interventional and quasi-experimental studies because

**Scoping review study on the prevalence and public health consequences of nonmedical use of tramadol in Africa**

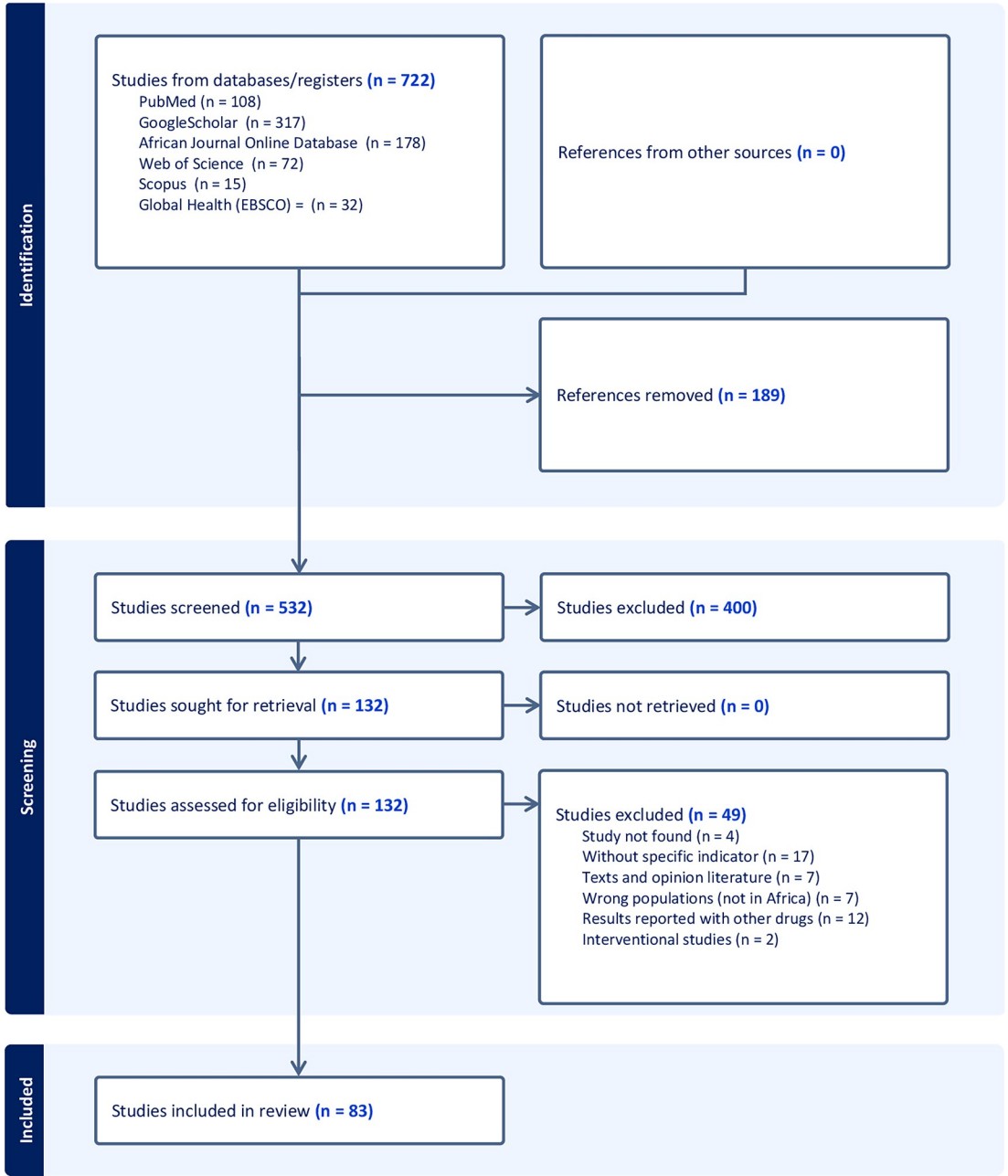

**Fig 1. PRISMA flow diagram of study selection.**

these studies involved NMU of tramadol exposure manipulation. Also, our scoping review did not include studies done in vitro to test tramadol addiction or toxicity; 3) studies in which tramadol is reported in combination with other drugs in such a way that it is not possible to specify isolated effects of tramadol; and 4) Texts and opinion literature.

**Stage 4: Charting the data.** Two researchers conducted data extraction on the Covidence platform and exported the data in Excel format, which included the author's name, publication year, study country, target population, study type, identified risk factors, and key findings related to the prevalence and health consequences of NMU of tramadol.

**Stage 5: Collating, summarizing, and reporting the results.** The quality of the included studies was not assessed [29, 32, 34]. The prevalence of tramadol use was reported by population categories grouped into five groups: 1) young adult/active populations; 2) professional group; 3) patient group; 4) academic group; and 5) others.

Health consequences were grouped using the WHO's International Classification of Diseases-11 (ICD-11). We identified four categories of health issues that can arise from the NMU of tramadol. The first category is intoxication, which refers to the changes in consciousness, perception, behaviour, and other physiological functions that can occur after taking tramadol. These effects are directly related to the drug's pharmacological properties and usually subside over time unless there are medical complications. The second category is dependence syndrome, which encompasses a range of cognitive, behavioural, and physiological symptoms that can develop after prolonged NMU of tramadol. These symptoms may include a strong desire to use tramadol, difficulty controlling its use, continuing to use it despite adverse consequences, and prioritizing tramadol over other aspects of life. The third category is withdrawal symptoms, which can be severe and varied and occur after stopping or reducing tramadol use. The onset and duration of these symptoms depend on the dose used before stopping or reducing use. The final category includes symptoms that cannot be classified into the three previously identified categories.

## Results

Our scoping review included 83 studies conducted between 2012 and 2023. Thirty-six studies were conducted in Egypt and 28 in Nigeria, 77% in the two countries. The remaining studies were distributed: 2 in Benin, Niger, Cameroon, and 7 in Ghana. Most studies were cross-sectional (60 studies), six were qualitative, and 2 were case studies. For more comprehensive characteristics of the studies included in our review, please refer to S3 Table.

### Subpopulations affected by NMU of tramadol

The populations affected by NMU of tramadol found in our scoping study can be divided into five categories: 1) the young/active population; 2) professional groups; 3) Patient Groups; 4) Academic groups; and 5) other groups. For further information on the classification of studies according to the populations studied, please refer to Table 1.

The first category includes young populations aged 10 to 35, young or active adults aged 15 to 55, teenagers, and young adults who are homeless [12, 35–40]. The second category includes professionals such as drivers and their assistants (motorcycle, tricycle, taxi, bus, private car, and truck) [41–52], workers in the construction and textile industries [42], skilled workers (hairstylists and mechanics) [53], hospital cleaners [54], and coffee shop workers [55]. The third category includes patients with tramadol dependence or opioid use disorders attributed to tramadol [56–65], patients with unintentional intoxication [66], patients with substance use disorders or drug addiction [67, 68], patients admitted to Neuro-Psychiatric Hospital [69], patients with seizures or psychotic episodes [70, 71], treatment-seeking men [72] or trauma

**Table 1. Classification of studies based on their type and the category of population examined.**

| | Addressing Prevalence and Health Consequences n (%) | Addressing Prevalence n (%) | Addressing Health Consequences n (%) | Total n (%) |
|---|---|---|---|---|
| Number of studies with Academic Group Population out of 83 selected studies | 0 | 20 (24.1) | 2 (2.4) | 22 (26.5) |
| Number of studies with other populations out of 83 selected studies | 0 | 5 (6.0) | 3 (3.6) | 8 (9.6) |
| Number of studies with Patient Group Population out of 83 selected studies | 3 (3.6) | 16 (19.3) | 18 (21.7) | 34 (41.0) |
| Number of studies with Professional Group Population out of 83 selected studies | 0 | 12 (14.5) | 3 (3.6) | 15 (18.1) |
| Number of studies with young adult/Active populations out of 83 selected studies | 0 | 6 (7.2) | 1 (1.2) | 7 (8.4) |

victims [73], adults attending family medicine center [74], pharmacy/dispensary customers [75], patients undergoing surgery [76] or admitted for acute myocardial infarction [77], women admitted to addiction treatment center [78], patients attending psychiatric clinic [79–82], and individuals presenting to poison control center for tramadol [83]. The fourth category pertains to students in both public and private schools in rural and urban areas, including secondary school students aged 13–18, upper secondary school students, and graduate or undergraduate university students [10, 72, 84–105]. The final group includes non-traditional populations, such as indigenous communities [17], teenagers attending summer camps [106], internally displaced [107] and various people [11, 18, 108–110].

According to the studies compiled in this review, there are various reasons cited for NMU of tramadol, which can be categorized into three reasons: 1) Physical and psychological reasons, 2) Social Reasons and 3) Economic Reasons. Physical and psychological reasons are factors related to a person's bodily function or psychological health. They refer to physical and psychological reasons, such as relief of pain, reducing stress, euphoria, protection against the cold, treatment for analgesic activities, combat fatigue, tiredness, staying awake, feeling more potent than usual, alleviating fear, pleasure-seeking and habits, prevention of withdrawal symptoms, craving or compulsive urges, mood improvement and increasing sexual performance (premature ejaculation/erectile dysfunction), and a sense of well-being [17, 18, 39, 43, 44, 46, 49, 55, 63, 72, 81, 89]. Social reasons are factors related to a person's social environment, including relationships, peer influences/pressure, influence of friends or partners, experimentation, novelty-seeking behaviour, and attempts to escape from troubles or stress in one's social environment [39, 43, 60, 81, 109]. Economic reasons relate to a person's financial situation or work life. They involve the need or desire to improve one's economic status, maintain or increase productivity at work, assist in continuing work, performance enhancement, have more motivation, or address other work-related issues [18, 39, 42, 46, 49, 89, 109]. They also include situations where someone takes tramadol because it is more affordable or accessible than other options.

## Risk factors associated with NMU of tramadol

Our scoping review identified the risk factors associated with the NMU of tramadol, which was distilled into four key categories: 1) age and residence: Tramadol nonmedical users tend to be younger and live in urban areas [47, 79, 86, 87, 111]; 2) socioeconomic factors such as irregular job status, a large family size, lower parental education levels, unmarried status, and insufficient monthly income strongly correlate with the NMU of tramadol [42, 46, 53, 84, 86]. Lack of parental control, familial tension, and having a friend or family member who uses

drugs are additional risk factors [46, 49]; 3) substance use and a history of abuse increase the risk of NMU of tramadol. Those who smoke or have a history of substance misuse are especially vulnerable, with a five times greater likelihood of misuse [39, 79, 89, 111]; 4) psychological factors, such as adverse childhood experiences, sociosexual behaviours and desires, and posttraumatic addiction, have been linked with NMU of tramadol [102].

## Prevalence of nonmedical use of tramadol

**Young/active population.** The NMU of tramadol within the youthful and active demographic exhibits significant variability prevalence, possibly indicative of divergent levels of exposure, risk, and accessibility among this specific population subset. A study involving adolescents revealed that out of 28 substance users, 10.7% engaged in NMU of tramadol [106] Similarly, within a group of substance users, a separate study indicated that among 29 individuals acknowledging substance abuse, a substantial portion (41.4%) reported tramadol consumption [40], suggesting that the NMU of tramadol might be intertwined with a broader trend of substance misuse. In Niger, 77.04% of homeless teenagers and young adults were identified as nonmedical tramadol users [12]. Among a study population ranging from ages 10 to 24, the prevalence of current tramadol usage was 6.6%, with even lower lifetime usage reported at 1.9% [38].

**Professional groups.** Some professionals use tramadol for nonmedical purposes to cope with the demands and stress of their jobs. Drivers (minibus, truck, motorcyclists, and bus) or professions, which often involve long hours and potential for fatigue, show a relatively high prevalence of NMU of tramadol, ranging from 7.2% to 35.1% [41, 42, 45–48, 52]. A high rate of 76% among commercial motorcyclists was found in one study, suggesting that these professionals may use tramadol to combat fatigue or handle work stress [51]. Construction workers, textile industry workers and rider businesses also show a high prevalence of 92.3%, 53% and 56%, respectively [42, 43]. Other Professional Groups (Hairstylists, Mechanics, and Coffee Shop Workers) show a relatively lower prevalence rate of 10.1%, 13.3%, and 13.1%, respectively [53, 55].

**Patient group.** The prevalence of NMU of tramadol also varies markedly across different patient groups. Among those with substance use disorders and opioid dependency, there is a high rate of NMU of tramadol, with prevalence rates at 77.1% to 92% [58, 62]. In psychiatric settings, a significant percentage of patients misuse tramadol, with 48.8% of those admitted to psychiatric hospitals and 46.9% of those attending psychiatric clinics reported [79, 82], and more than half of the patients (56.6%) presented to the Poison Control Center were tramadol nonmedical users [83].

Furthermore, over half (54.4%) of patients attending addiction clinics misuse tramadol non-medically, and 78% are addicts [81]. Patients admitted with acute myocardial infarction reported a 29% prevalence [77]. The lowest rates were observed with trauma patients, adults attending family medicine centers, females with psychoactive substance dependence, and consumers of psychotropic medicines, with prevalence rates ranging from 7.8% to 16.2% [73, 74, 80].

**Academic groups.** Again, the NMU of tramadol demonstrates a substantial variation across academic groups. For instance, a considerable rate of tramadol misuse was found among undergraduate student substance users (74.2%) [100] and secondary school substance abusers (83.3%) [98]. Three research in Egypt show that many students who use drugs also use tramadol for nonmedical reasons. The percentages are high: 54.7%, 31.1%, and 21% [10, 97].

Various studies have indicated that students with substance abuse problems have lower rates of NMU of tramadol, with prevalence rates of 18.9%, 12.9%, and 6.3%. [90, 94, 99]. The

lowest rates of NMU of tramadol were observed in Benin and Nigeria, where the prevalence rates among students with a history of substance abuse were 1.81% and 4.4%, respectively [88, 91].

The prevalence of NMU of tramadol among high school students differs notably across diverse academic settings and geographical regions. For instance, a significant prevalence rate of 31.0% was discovered among a group of 418 senior high school students [85]. Contrarily, another study reported a relatively lower prevalence of 12.1% within a general high school population [35]. Taking a broader geographical scope, in Cameroon, a study noted a lifetime prevalence of 7.5% among a sample of 625 high school students [96] and a rate of 10.79% was documented among 315 secondary school students [105] in Nigeria, while in Egypt, a study uncovered an 8.8% prevalence rate within a cohort of 204 students [86]. In West Africa, a study in Benin observed a prevalence of 9.6% within a diverse sample of 384 secondary school students. This study also revealed a significant gender-related discrepancy between male (13.4%) and female (4.4%) students [89].

Investigations into the NMU of tramadol among university students have also yielded varied prevalence rates across distinct study populations and regions. A study conducted in Egypt reported a prevalence rate of 5.7% within a sample of 283 university students [84]. This figure contrasts notably with a more extensive study in Egypt, which found a prevalence of 12.3% among 1173 university students [10]. In a considerably broader survey in the same country, another analyzed tramadol use across a large student population of 7445, revealing a lower overall prevalence rate of 3% [93]. Similarly, two other studies show a low prevalence of tramadol misuse among diverse student populations. One study found a rate of 1% [92], while another showed 1.8% among a sample of 500 students [95]. Contrasting with these lower prevalence rates, a study in Nigeria observed a high prevalence of 35.0% within a sample of 400 undergraduate students [101].

**Other groups.**   There are varying rates of NMU of tramadol among different groups. In a survey of 150 pharmacy patients conducted in Mali, the prevalence was 11.3%, while a survey of 30 pharmacists showed that, from their point of view, the prevalence of NMU of tramadol by their patients was 10.8% [75]. Additionally, 38 internally displaced substance users in Nigeria found a prevalence of 10.5% [107]. Another survey of teenagers in summer camps showed a prevalence of 10.7% among substance users [106].

## Health consequences of nonmedical use of tramadol

**Intoxication.**   Symptoms in this category include loss of consciousness, incoherent and irrational speech, and altered consciousness levels [11, 109]. Physiological symptoms commonly associated with the disease include seizures, headaches, dizziness, nausea, vomiting, muscle pain, abdominal cramps, and skin rash [17, 36, 44, 83]. A study conducted in Togo found that after ingesting tramadol, convulsions occurred in 33.3% of cases, headaches in 60.3%, dizziness in 62.2%, skin rash in 67.6%, muscle aches in 54.9%, stomach cramps in 60.4%, and nausea/vomiting in 58.9% [44]. In Egypt, a study found that tramadol-induced seizures represented 7% of all patients and 12% of male patients [70]. Other manifestations of intoxication include behavioural disturbances such as restlessness, irritability, destructive behaviour, and hyperactivity [11, 57, 109].

**Dependence syndrome.**   Studies have demonstrated an intense desire among users to continue using tramadol for pain avoidance, pleasure-seeking, and habit-forming purposes [72]. Additionally, NMU of tramadol can lead to risky behaviours, eating disorders, social stigma, general feelings of discomfort, sadness, loss of interest or pleasure and lack of respect [49, 56, 62, 112]. NMU of tramadol increases the severity of nicotine dependence, and the relation is

bi-directional [65]. The finding also reveals a negative relationship between the NMU of tramadol and students' disruptive behaviour among secondary students (r = -0.228, p = 0.033) [104]. Tramadol addiction can cause brain changes, leading to riskier decisions and increased impulsivity [61]. Moreover, NMU of tramadol can result in mild to severe drug use disorders. A study found that nearly half of nonmedical users had psychiatric disorders, with mood disorders being the most prevalent at 59.2%, anxiety disorders at 38.8% and psychotic disorders at 26.5%. Borderline and antisocial personality disorders were the most common comorbid personality disorders found [111].

**Withdrawal syndrome.**   These symptoms can include sleep disturbances such as insomnia [60, 110], physical symptoms like severe pain, craving and fatigue, palpitations, and tremors [110], as well as emotional and behavioural symptoms such as irritability, anxiety, depression, attention disorders and emotional aloofness [50, 63, 109].

**Other symptoms.**   Several other consequences of NMU of tramadol do not neatly fit into these categories but have been reported in our scoping review findings. These include the negative impact on cognitive performance [64], decreased sexual self-esteem and increased sexual depression and preoccupation [56], a significant association with risky sexual behaviours [62], and potential impairment in the reabsorption function of nephrons [67]. Additionally, a significant correlation has been found between the NMU of tramadol and reduced life value moderated by moral identity [112]. According to Maiga et al. (2013), tramadol misuse has been associated with organized fights, kidnappings and rapes, murders, and traffic accidents [18]. In addition, tramadol misuse has been independently associated with adverse cardiovascular effects [77].

## Discussion

### Significance of findings

Our scoping review has uncovered a previously underexplored phenomenon: the NMU of tramadol in African countries. We have identified five distinct populations affected by this phenomenon—young/active individuals, professional groups, patient groups, academic groups, and others. These populations experience various effects, including intoxication, dependency-like syndromes, withdrawal symptoms, and other related manifestations. The risk factors for NMU of tramadol can be grouped into four primary categories: age and residency, socioeconomic conditions, history of substance abuse, and psychological factors. Notably, prevalence rates differ among these groups. For example, there is a pronounced use among students and professionals, particularly among transportation operators such as taxi, bus, minibus, truck, and motorcycle drivers. The reasons for NMU of tramadol can be divided into three main categories: physical and psychological reasons, societal influences, and economic pressures. The patterns of misuse vary, possibly influenced by factors such as stress, peer pressure, and ease of access to the drug. Our study underscores the urgent need for targeted interventions, particularly in educational settings identified as hotspots for substance misuse initiation. Furthermore, it also highlights the necessity of understanding the context of the NMU of tramadol in different occupational settings. Finally, our findings stress the importance of further extensive research into the NMU of tramadol to develop effective public health strategies and policies.

### Comparative literature

Our findings relating to the population groups primarily involved in the NMU of tramadol are similar to other studies conducted in other world regions. For example, a meta-analysis conducted in Iran showed that young people aged 15–34, university and high school students, male prisoners incarcerated, illicit drug users, and individuals using injectable drugs were

mainly involved in the NMU of tramadol [26]. Male university students had a lifetime prevalence of 5–17.8%, while females had a lower rate of 0–7.9%. In high school students, 7.8% of males and 1.8% of females reported lifetime use. The meta-analysis also documented tramadol-related complications such as poisoning, seizures, and fatalities, similar to our findings [26].

In contrast to our findings, another review found that NMU of tramadol was relatively infrequent compared to other drugs like codeine, morphine, and oxycodone, registering the lowest or second-lowest rates of misuse in all surveyed countries such as Germany, Italy, Spain, and the United Kingdom [27]. Incidences of NMU of tramadol ranged from 7.27 cases per 100,000 units sold in Germany to 54.92 cases in the United Kingdom [27]. This study further reinforces the notion that tramadol is sometimes called the "cocaine of the poor" [14]. As noted, there are distinct differences between the opioid crisis in North America, predominantly driven by methadone and fentanyl, and the situation in Africa with tramadol [21]. In Africa, the crisis is marked by an almost complete absence of medical tramadol prescriptions, and the most used tramadol came from illicit trafficking, unlike the formal channels prevalent in North America. The root causes also differ. While North America's opioid use largely stems from pain management, in Africa, it is majorly influenced by socioeconomic conditions [21].

The other finding of our study is the relative recency of the included studies. The oldest study was conducted in 2012, corresponding to the year in which another painkiller, dextropropoxyphene, was discontinued in European markets, followed by African markets, due to the risk of toxicity or overdose [113]. Indeed, on June 14, 2010, the European Commission confirmed the European Medicines Agency's unfavourable opinion on maintaining authorizations for specialties containing dextropropoxyphene. This decision would be followed 15 months later by withdrawing specialties containing the said molecule [113]. According to a study, the increase in prescribing and consumption of tramadol is a direct consequence of this discontinuation [9]. This study anticipated an increase in the NMU of tramadol in low- and middle-income countries, as already observed in Iran. Given the current situation of tramadol consumption in African countries, it would be wise to question this decision [9].

Also, most studies (77%) were conducted in Nigeria and Egypt. This could be due to various reasons, such as the fact that these two countries are among the most populated in Africa, which could make the illegal tramadol market more profitable for traffickers. This increase in the NMU of tramadol may have led to more research interest in these countries. Also, differences in how the body processes tramadol may vary between countries due to genetic variations. Tramadol is metabolized by the liver enzyme CYP2D6, which produces a more potent form, M1, responsible for tramadol's pain-relieving effects [1]. However, genetic variations in the CYP2D6 enzyme can significantly affect tramadol's effectiveness and safety [114, 115]. There are four categories of metabolizers based on genetic variations [116]: ultra-rapid, normal, intermediate, and poor. Each category affects how efficiently tramadol is metabolized. Ultra-rapid metabolizers are at risk for harmful effects; normal metabolizers metabolize efficiently, intermediate metabolizers may have decreased pain relief, and poor metabolizers experience reduced pain relief [114]. Metabolizer phenotypes vary by race and ethnicity [116]. Poor metabolizers are 5–10% in Caucasians but less common in Asians and Africans. Ultrafast metabolizers are more common in North African, Middle Eastern, and Oceanic populations and less common in Europeans and Asians [114, 116]. These genetic differences in metabolism may also explain the increased NMU of tramadol in the countries mentioned earlier, particularly Egypt and explain differences in tramadol response.

Our findings also reveal a significant misuse of tramadol among academics. Various factors may contribute to this trend, including the intense pressure associated with academic performance, peer influence, and emotional or social challenges from these developmental periods.

Some students turn to substance abuse as a maladaptive coping mechanism. Social pressure, especially on adolescents and young adults, has long been recognized as a significant factor in drug use. Additionally, the strength of the young person's relationships with primary sources of socialization (family and school) plays a crucial role in determining the effectiveness of norm transmission [117]. Although any socialization link can transmit deviant norms, healthy family and school systems are more likely to transmit prosocial norms [117]. Peer groups can also transmit prosocial or deviant norms, but the primary source of deviant norms is the peer group [117]. Weak ties between family and child and/or school and child increase the likelihood of associating with a deviant peer group and adopting deviant behaviours [117]. Similarly, weak peer bonds can also increase the risk of bonding with deviant peers. These findings support the primary socialization theory, offer insights for improving prevention and treatment, and suggest the need for further research.

## Implications for practice, public health and policy

The results of our study have critical implications for practice, public health, and policy. Practically speaking, it is essential for health professionals, especially primary care providers, pharmacists and physicians, to be better informed about the trends and population groups associated with the NMU of tramadol. This information could guide tailored medication counselling for population groups susceptible to tramadol misuse when they are in a clinical environment seeking treatment that necessitates tramadol prescription. From a public health perspective, there is an urgent need to launch extensive awareness campaigns about the dangers of NMU of tramadol, mainly aimed at high-risk groups such as students and drivers is crucial. Systematic monitoring could offer insights into evolving trends and assist in efficiently allocating resources for prevention and treatment in vulnerable communities. On the policy front, implementing stricter control mechanisms, improved border checks, and harmonizing sub-regional and continental pharmaceutical regulations could help curtail illicit trafficking. Further, research funding could be directed toward understanding the long-term effects of the NMU of tramadol and designing effective interventions with an integrated approach that fosters collaboration between sectors such as health, education, social sciences, transport, and law enforcement.

## Strengths and limitations

Our study also pinpointed several areas where further research on the NMU of tramadol is warranted. Considering economic, sociological, and cultural factors, it is vital to delve deeply into the mechanisms and motivations behind the NMU of tramadol. Moreover, there is a need to investigate further the broader economic implications of the NMU of tramadol, ranging from workforce productivity losses to healthcare expenses and societal burdens. Longitudinal studies are required to uncover long-term impacts on the health, economy, and safety of countries affected by this phenomenon. Comparative studies could shed light by comparing NMU of tramadol across different regions or countries, highlighting regional influences or policy outcomes. The effectiveness of current awareness campaigns could be further analyzed to refine future public education initiatives. Probing into these domains will significantly contribute to formulating measures to address the mounting challenges NMU of tramadol presents.

Despite all the measures taken, our scope review has several limitations. The prevalence of NMU of tramadol is estimated in various ways, using different assessment methods that produce varying percentages. Assessment methods range from self-reporting to more objective techniques like urine testing and gas chromatography. These different approaches lead to significant differences in reported prevalence rates. Furthermore, studies may focus on "lifetime"

use, including any NMU of tramadol throughout an individual's life, or "current" use, referring to use within a specific recent period like the past month or year. Moreover, as we have observed, the studies in our scoping review focus on different population types, complicating the direct comparison of the results. Further adding to this complexity is the variation in how prevalence rates are reported. Sometimes, the prevalence is reported for the entire study population, while often, the prevalence of tramadol consumption is expressed concerning a subpopulation of drug users. This heterogeneity of study populations and reporting methods underlines the challenges in forming a comprehensive and unified understanding of the prevalence of the NMU of tramadol.

Despite these limitations, our scoping review is the first conducted in Africa on the NMU of tramadol. It has several significant strengths, including comprehensive geographical coverage, encompassing multiple African countries, and providing an overarching view of the situation on the continent. The included studies enabled the potential identification of evolving trends and changes over time. Incorporating various study designs, from cross-sectional to qualitative research and case studies, facilitates a multifaceted understanding of the topic. Our findings also helped to clarify the specific populations affected by the phenomenon, paving the way for tailor-made interventions and policy initiatives. By identifying and categorizing key risk factors, it becomes feasible to formulate preventive strategies and adapt interventions to particularly vulnerable groups. Additionally, examining prevalence rates across different groups and the health repercussions of the NMU of tramadol offer a comprehensive picture of the problem's magnitude, underscoring the necessity for timely intervention.

## Conclusion

Our scoping review of 83 studies conducted between 2012 and 2023 offers essential information on the NMU of tramadol in several African countries. Our results showed that the populations most affected are diverse, ranging from young people to professional groups, patients, academic communities, and others. The prevalence of NMU varies considerably from one sub-population to another, underscoring the complexity and multifaceted nature of tramadol misuse. The variability of these rates in similar demographic groups across studies and regions is striking.

NMU of tramadol is a multidimensional issue with far-reaching economic, societal and safety implications. Thus, this study highlights the need for more nuanced and targeted interventions that address the unique challenges and factors contributing to tramadol misuse in different African demographics and regions. It is also important to note that many factors contribute to the well-being of populations, such as the political and legislative context, the economic context, educational systems, social services, employment support, family life and the housing environment. Therefore, to fully understand the consequences of the NMU of tramadol, future studies should focus on studying the social and economic cost of this abuse on households. This will enable us to better understand the impact of health determinants on household well-being.

## Supporting information

**S1 Checklist. Preferred Reporting Items for Systematic reviews and Meta-Analyses extension for Scoping Reviews (PRISMA-ScR) checklist.**
(DOCX)

**S1 Table. PICOS search criteria and sources for the review.**
(DOCX)

**S2 Table. Search strategy.**
(DOCX)

**S3 Table. Characteristics of included studies.**
(DOCX)

## Author Contributions

**Conceptualization:** Saidou Sabi Boun, Sanni Yaya.

**Data curation:** Saidou Sabi Boun.

**Formal analysis:** Saidou Sabi Boun.

**Investigation:** Sanni Yaya.

**Methodology:** Saidou Sabi Boun, Olumuyiwa Omonaiye, Sanni Yaya.

**Project administration:** Olumuyiwa Omonaiye, Sanni Yaya.

**Supervision:** Olumuyiwa Omonaiye, Sanni Yaya.

**Validation:** Sanni Yaya.

**Writing – original draft:** Saidou Sabi Boun.

**Writing – review & editing:** Olumuyiwa Omonaiye, Sanni Yaya.

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
