## [Decision Letter · Decision Letter 0]

10 Nov 2023

PGPH-D-23-01629

Prevalence and health consequences of nonmedical use of Tramadol in Africa: A systematic scoping review

Dear Dr. Yaya,

Thank you for submitting your manuscript to PLOS Global Public Health. After careful consideration, we feel that it has merit but does not fully meet PLOS Global Public Health’s publication criteria as it currently stands. Therefore, we invite you to submit a revised version of the manuscript that addresses the points raised during the review process.

Your manuscript has been evaluated by two reviewers, and their comments are appended below.

Reviewer 1 is satisfied with your manuscript in its current form; Reviewer 2 has requested some clarifications and has provided some suggestions for revisions. Please ensure you address each of the reviewers' comments when revising your manuscript.

We look forward to receiving your revised manuscript.

Kind regards,

Hugh Cowley

Staff Editor

Journal Requirements:

1. Please also ensure all files are under our size limit of 10MB.

Additional Editor Comments (if provided):

Reviewers' comments:

Reviewer's Responses to Questions

**Comments to the Author**

1. Does this manuscript meet PLOS Global Public Health’s publication criteria? Is the manuscript technically sound, and do the data support the conclusions? The manuscript must describe methodologically and ethically rigorous research with conclusions that are appropriately drawn based on the data presented.

Reviewer #1: Yes

Reviewer #2: Yes

2. Has the statistical analysis been performed appropriately and rigorously?

Reviewer #1: Yes

Reviewer #2: Yes

3. Have the authors made all data underlying the findings in their manuscript fully available (please refer to the Data Availability Statement at the start of the manuscript PDF file)?

Reviewer #1: Yes

Reviewer #2: Yes

4. Is the manuscript presented in an intelligible fashion and written in standard English?

Reviewer #1: Yes

Reviewer #2: Yes

5. Review Comments to the Author

Reviewer #1: The manuscript is worth publishing and I request the manuscript to be accepted. The authors have made an good attempt in writing the manuscript and the misuse of tramadol in the African countries and it is worth publishing.

Reviewer #2: Review Comments on Manuscript PGPH-D-23-01629: Prevalence and Health Consequences of Non-medical Use of Tramadol in Africa: A systematic Scoping Review

This is a well-researched scoping review. I applaud the efforts of the authors to put together a “pooled” prevalence of non-medical use of tramadol in Africa which is a continent most hit by opioid crisis in contemporary times. However, I believe the review manuscript would be made richer if the authors consider the following comments:

Abstract

1. The abstract should cover in statistical terms, the prevalence of non-medical use of tramadol in Africa, highlighting the percentages for each non-medical use as reported in the studies reviewed.

2. The prevalence (in percentages) of NMU of tramadol among young adults, professionals, patients and academics should be stated.

Introduction

1. Paragraph 1: Mention those countries where tramadol is put under control (especially in Africa).

2. Paragraph 1: Define non-medical use of tramadol.

3. Paragraph 3: mention some common brand names of tramadol in African countries.

4. Has there been any review study on tramadol in Africa or elsewhere? What aspects have been covered and what is the gap for this review study to fill?

Results

1. The section on “Prevalence of Non-medical Use of Tramadol” should be presented before the section on “Health Consequences of Non-medical Use of Tramadol”.

Discussion

1. Under “Comparative literature”, paragraph 2: first line: different reference style used here: Iwanicki et al. (2020).

2. Third paragraph under “comparative literature” looks better if presented in the “Introduction”.

Conclusion

1. The conclusion is too lengthy. It should cover only the salient points drawn from the findings of the review study. It shouldn’t look like another “discussion” of findings.

Reviewer: Dr. Orfega Zwawua

6. PLOS authors have the option to publish the peer review history of their article (what does this mean?). If published, this will include your full peer review and any attached files.

**Do you want your identity to be public for this peer review?** For information about this choice, including consent withdrawal, please see our Privacy Policy.

Reviewer #1: **Yes: **Prof Mayur Yergeri Chandrasekharappa

Reviewer #2: **Yes: **Dr. Orfega Zwawua

---

## [Decision Letter · Decision Letter 1]

15 Dec 2023

Prevalence and health consequences of nonmedical use of Tramadol in Africa: A systematic scoping review

PGPH-D-23-01629R1

Dear Dr. Yaya,

We are pleased to inform you that your manuscript 'Prevalence and health consequences of nonmedical use of Tramadol in Africa: A systematic scoping review' has been provisionally accepted for publication in PLOS Global Public Health.

Best regards,

Julia Robinson

Executive Editor

Reviewer Comments (if any, and for reference):

Reviewer's Responses to Questions

**Comments to the Author**

1. If the authors have adequately addressed your comments raised in a previous round of review and you feel that this manuscript is now acceptable for publication, you may indicate that here to bypass the “Comments to the Author” section, enter your conflict of interest statement in the “Confidential to Editor” section, and submit your "Accept" recommendation.

Reviewer #2: All comments have been addressed

2. Does this manuscript meet PLOS Global Public Health’s publication criteria? Is the manuscript technically sound, and do the data support the conclusions? The manuscript must describe methodologically and ethically rigorous research with conclusions that are appropriately drawn based on the data presented.

Reviewer #2: Yes

3. Has the statistical analysis been performed appropriately and rigorously?

Reviewer #2: Yes

4. Have the authors made all data underlying the findings in their manuscript fully available (please refer to the Data Availability Statement at the start of the manuscript PDF file)?

Reviewer #2: Yes

5. Is the manuscript presented in an intelligible fashion and written in standard English?

Reviewer #2: Yes

6. Review Comments to the Author

Reviewer #2: The manuscript is worth publishing now. However, authors should check References Numbers 49, 73, 103 and 111 and change the upper cases to lower cases.

7. PLOS authors have the option to publish the peer review history of their article (what does this mean?). If published, this will include your full peer review and any attached files.

**Do you want your identity to be public for this peer review?** For information about this choice, including consent withdrawal, please see our Privacy Policy.

Reviewer #2: **Yes: **Dr. Orfega Zwawua
